# The Influence of Street Components on Age Diversity: A Case Study on a Living Street in Shanghai

**Dadi An** [1], **Yan Liu** [1] and **Yihua Huang** [2,*]

[1]    School of Art Design and Media, East China University of Science and Technology, Meilong Road No.130, Shanghai 200237, China; andadi@ecust.edu.cn (D.A.); liuyan991121@126.com (Y.L.)
[2]    Shanghai Academy of Fine Arts, Shanghai University, Shangda Road No.99, Shanghai 200444, China
*    Correspondence: yihua_huang@shu.edu.cn

**Abstract:** Background: Living streets are vital spaces for urban residents to socialise and enjoy leisure activities. Previous research has examined the relationship between street components and residents' public lives but has overlooked the age composition of residents. To move towards all-age-friendly spaces, it is essential to investigate the preferences of different age groups for public life and to enhance the age diversity of cities. Methods: Based on a literature review, this research selected an index system that consists of four domains (street space, street facilities, street layout, and commercial features) and was suitable for our survey site. The indicators were chosen according to their relevance to street components and measurability. This research then conducted offline surveys and observations to assess the components of small-scale public places on Golden Street in Shanghai. Results: Data analysis showed that street facilities and commercial features strongly correlate with age diversity. However, some indicators did not yield the expected results. Conclusions: Our research demonstrates that specific street components are strongly associated with age diversity. Optimising these components may improve the age diversity of cities. The conclusions suggest that future street construction projects be implemented to assist relevant management departments in creating dynamic streets.

**Keywords:** street space; age diversity; public life; quantitative analysis

## 1. Introduction

The World Health Organisation (WHO) introduced a framework for creating age-friendly cities and communities in 2007. This framework was intended to improve the physical and social environment for residents of all ages through the design of policies and services that support their needs. According to China's 2021 national demographic census, the ageing population was rapidly growing, and the proportion of children was also increasing because of the two-child policy. These situations presented challenges for generational coexistence. The government was advocating stock planning and had proposed micro-renewal concepts to address existing urban planning issues. The objective was to enhance age diversity in cities by addressing daily challenges and creating healthy, comfortable, and liveable urban environments.

As globalisation and urbanisation continue to accelerate, demand is growing for space in people's daily lives and work. The flow of people in certain areas has increased significantly. However, the uncontrolled expansion of urban construction often overlooks residents' feelings and requirements, leading to many problems [1]. Thus, the focus of urban renewal is now on a city's streets, which are its lifelines. Streets are versatile public spaces that embody society, economy, and culture. They are standardised places for human movement with economic and social implications beyond physical identity [2]. Streets are the smallest units for scrutinising society and virtual spaces for public life [3]. Prioritising human vitality and spatial factors is therefore crucial [4–6].

The streets are home to a diverse range of activities. The elderly often stroll with their grandchildren, parents pick up their children from school and shop together, and people of various ages move around simultaneously. It is common to see the elderly exercising in the morning, young people eating at noon, and children roller skating in the afternoon, all in the same space. All-age space is becoming increasingly prevalent, and it has fostered extensive academic debate. However, there is a lack of consensus on the definition of 'all-age-friendly', as well as quantitative analysis [7], although Liu's research defined age diversity as the degree of crowd mixing [8]. In addition, few discussions existed about the manifestation of all-age diversity, including simultaneous and staggered sharing.

Therefore, to promote the micro-renewal of street space to create all-age-friendly spaces, our research aimed to achieve the following three objectives:

- To understand the quantification of age diversity and identify the relationships between street components and age diversity to identify critical elements.
- To focus on the critical elements to explain what kind of public space is most suitable for co-living and how space transformation can mitigate coexistence issues between the old and young.
- To provide insights for urban micro-renewal and offer valuable recommendations for revitalising streets and making cities more vibrant.

## 2. Review

Previous studies had primarily considered age diversity using two approaches. The first was conceptual dissection, which had received the most attention. Scholars had defined an all-age-friendly space as an environment that welcomes people of all ages and supports them to share material space, knowledge, and skills. A supportive, inclusive, and barrier-free material space environment that matches the vitality, stability, and health of residents of different ages was necessary [7]. The second approach, quantitative analysis, has been largely neglected in the literature. Although some scholars had conducted correlation analyses by dividing the population into categories [9,10], few studies had considered convergence indicators of age diversity. Therefore, our research aimed to quantify age diversity and its relationship with spatial components. The literature review in this study was conducted from four perspectives, and a detailed analysis is presented in Tables 1 and 2.

**Table 1.** Summary of relevant foreign research.

| | Author | Mohadeseh Mahmoudi [11] | Madeleine Steinmetz-Wood [12] | Shadi Zang Zarin [13] | Youn-Won Kang [14] |
|---|---|---|---|---|---|
| | Dependent Variable Selection | Quality and liveability | Walking level | Relevance and availability, facility diversity, benefits, etc. | Importance and satisfaction |
| Street Spatial Features | Street Length and Width | | √ * | | √ |
| | Traffic Accessibility | √ | √ | √ | √ |
| | Facade Characteristics | | | √ | √ |
| Street Furnishings | Garbage Can Density | | | √ | |
| | Rest Seat Density | √ | √ | √ | √ |
| | Green View Rate | √ | | √ | √ |
| Street Layout Features | Sky Openness | | | √ | |
| | Flat Pavement | √ | | √ | √ |
| | Colour Aesthetic Characteristics | √ | √ | √ | √ |
| Commercial Features | Store Density | √ | | | |
| | Business Mix | | | | |
| | Social Places Density | | | | |
| | Surrounding Dwelling Density | | | √ | |
| | Data Acquisition | Questionnaire | Google Street View | Field research, questionnaire survey | Field research |
| | Technical Method | Correlation analysis | Virtual-STEPS | Correlation analysis, regression analysis | IPA analysis |
| | Research Area | Street (2) | Community (40) | Street (2) | Street (2) |

* √ means the author used the corresponding indicator of the line.

**Table 2.** Summary of relevant domestic research.

| | Author | Huang Dan [15] | Liu Song [8] | Chen Yong [16] | Hou Yunjing [17] | Long Ying [10] | Jiang Lei [18] |
|---|---|---|---|---|---|---|---|
| | Dependent Variable Selection | Number of residents, time-age structure, and type of behaviour | Demographic density and age diversity | Stay activity density | Recreational physical activity capacity | Intensity of tourists' walking and parking behaviour | Number and length of stays |
| Street Spatial Features | Street Length and Width | √ * | | √ | √ | √ | |
| | Traffic Accessibility | | √ | | | | √ |
| | Facade Characteristics | √ | | √ | | √ | |
| Street Furnishings | Garbage Can Density | | | | | | √ |
| | Rest Seat Density | √ | | | √ | √ | √ |
| | Green View Rate | √ | √ | | √ | √ | √ |
| Street Layout Features | Sky Openness | | √ | | √ | | |
| | Flat Pavement | √ | | | √ | √ | √ |
| | Colour Aesthetic Characteristics | | | | | | |
| Commercial Features | Store Density | √ | √ | √ | | | √ |
| | Business Mix | √ | | √ | | | √ |
| | Social Places Density | √ | | | | | √ |
| | Surrounding Dwelling Density | | √ | | | | √ |
| Organisational Features | Structural Topological Relationship | | | | √ | | |
| | Data Acquisition | Field research | Cell phone signalling | Field research | Network research, visual analysis | Field research, machine algorithms | On-site observation, online inquiry |
| | Technical Method | Linear regression, one-way variance | Correlation analysis, linear regression | Multiple logistic regression | Space syntax | Linear regression | Correlation analysis |
| | Research Area | Street (19) | Waterfront Space (85) | Street Section (17) | Park (16) | Street (144) | Street (8) |

* √ means the author used the corresponding indicator of the line.

### 2.1. Perspective

As shown in Tables 1 and 2, streets could be examined from the macro, meso, and micro perspectives. Scholars adopting a macro perspective observed street components from an urban planning standpoint and utilised data to analyse the impact of multiple streets on a city's overall performance [10,19]. Streets often adhere to general planning guidelines and represent the smallest component of a city. In contrast, the meso perspective focused on comparing several streets of similar types to interpret their intended functions based on differences [11,14,18]. Finally, the micro perspective frequently entailed the selection of small places for field observation and concentrating on one or two streets to investigate public life and micro-spaces [13,16,20]. For example, Zhang (2022) utilised field investigation, street mapping, data statistics, and other methods to conduct on-site observations of Shantang Street in Suzhou [21]. Fang (2021) used Tianzifang in Shanghai to develop a model for measuring street attributes [22].

Hence, this study examined small-scale public spaces by observing the facilities and layout from a pedestrian's viewpoint [23]. This research explored the correlation between street space components and crowds to gain insights into the design and functionality of these areas.

### 2.2. Environment Indicators

The street was a multifaceted entity that required considerable effort in the initial stages of research to identify objects and classes, attributes and operations, and relationships to use object-oriented methodologies effectively [24]. A quantitative index system for small-scale public places could be established based on the literature, and a street's elements were usually divided into four categories: street space, street facilities, street layout, and commercial features. The selection and application of these indexes were interpreted from these four aspects, in conjunction with Tables 1 and 2.

(1) Street space usually referred to the physical components of the street created by architecture and zoning and included the street's length and width [13–16], architectural features [13–15,25] (indicators generated by street architecture usually included facade continuity, facade sticker rate, and facade transparency), and traffic accessibility [12,13,18] (whether people could reach their destinations quickly and safely).

(2) Street layout included sky openness [8,13] (usually determined by the proportions of buildings and streets, which affected people's view), flatness of the pavement [11,13,14,17,18] (whether people could easily walk, which was related to the possibility of danger), colour aesthetic characteristics [13,14], and street connectivity [26] (the opportunity and convenience of coming to the street).

(3) Street facilities refer to infrastructure on the street for people's daily activities, including seats to rest [11,12,14,15,27,28], garbage cans [12,14,17], and the green view rate [12,14,16].

(4) Commercial features include commercial stores [8,27,29], social places [14], and surrounding residential areas [12,17], and are calculated in terms of quantity statistics, density, and diversity.

Research based on quantitative standards was comprehensive and replicable. Nevertheless, each research domain must use distinct indicators. The selection of measurement indicators should account for the particularities of the target site and align with field-specific considerations.

To address this issue, this research chose a survey site, delineated the spatial component indicators, excluded unsuitable indicators, and incorporated variables that may influence age diversity. The components were categorised into four groups to comprehensively capture the indispensability of various street space components.

### 2.3. Demographic Index

Numerous studies have aimed to quantify the relationship between crowd attributes and spatial components, although methodologies vary and few studies have used age as a determining factor. Scholars typically scrutinise crowd activities and emotions.

Mehta (2007) examined people's social activities [30], and Wineman et al. (2014) investigated walking behaviour in different neighbourhoods [31]. Emotional factors and social activities had also been within the scope of research [14,27,32,33]. In recent years, numerous regions have introduced the concept of age-friendliness, and Shanghai has vigorously promoted the 15-min community-life circle as a strategic deployment for the elderly, which brings residents within a 15-min walk of services. As such, recent research has centred more on the elderly, aligning with policy on ageing [34]. Furthermore, some studies have concentrated on creating child-friendly cities [35]. Research on subdivided groups classified the groups into categories [9,10].

However, there was a lack of research on age diversity in public spaces, and studies on the topic had primarily focused on the overall population with limited attention given to demographic subgroups [36,37]. Furthermore, population bias may exist in previous studies, as public streets are complex spaces that involve individuals of all ages. Therefore, this research proposed a classification method that clustered individuals by age group, dividing the crowd into four categories, and used the index of age diversity to evaluate the degree of mixing of all groups using the entropy method [8]. Despite the potential benefits of a street designed to promote all-age friendliness, there has been limited investigation into two critical aspects of this concept: simultaneous and staggered sharing. This research integrated the two forms, as both could foster all-age friendliness on public streets. Simultaneous sharing could promote intergenerational communication, and staggered sharing could create a flexible and efficient use of space.

### 2.4. Method

Scholars had proposed various data acquisition and analysis methods, which can be broadly classified into two categories. The first approach involved using big data to collect significant user data for analysis [11–15,29]. For example, Li (2022) utilised many street-view pictures in their research [38]. The advantage of big data lies in its ability to provide sufficient data support and its considerable analytical capabilities. However, its broad application range must be used in conjunction with actual case analysis. The second method involved acquiring small amounts of data through field research [14,15]. In Ozbil's research (2019), he explored the relationship between street characteristics and pedestrian flows [27]. Field research was the most direct method of observing a research site and could also be used to identify problems from a pedestrian's perspective. Therefore, this research conducted field surveys, took photos, and captured videos to obtain detailed data on the indicators.

Regarding the methods for data analysis, previous studies had used correlation analysis [8,11,13,18], linear regression analysis [8,15,38], and stepwise regression analysis [8]. Linear regression analysis had been the most commonly used. Despite being a suitable method, stepwise regression analysis has been criticised by some scholars because of its limitations [29]. Thus, this research used correlation analysis for the initial examination and integrated it with linear regression analysis for further data processing.

### 2.5. Summary

The gaps in existing theory are mainly related to the following three points:

- The practise of micro-updating at the micro-scale level was not sufficient;
- The methods and framework for quantifying street space indicators were not comprehensive;
- In the design of demographic factors, age was not considered in sufficient depth, and the perspective of age diversity was lacking.

This research aimed to establish correlations between age and street components by quantifying age diversity. This approach offered urban designers and planners a more intuitive means of understanding the living conditions of street residents and enhancing neighbourhood environments. To achieve these objectives, our research conducted a comprehensive literature review to design a street quality index system and used field observation to collect data. Subsequently, data analysis was performed to interpret the regional age diversity and achieve the research objectives.

## 3. Research Design

This study explored the correlation between age diversity and street spatial components, so the research design was divided into three steps, as shown in Figure 1. The first step was to determine the indicator system and survey scope in the early stages; the second was to obtain sample data through field research; and the third was to conduct thermal maps, correlation analysis, and regression analysis of the data to obtain the correlations.

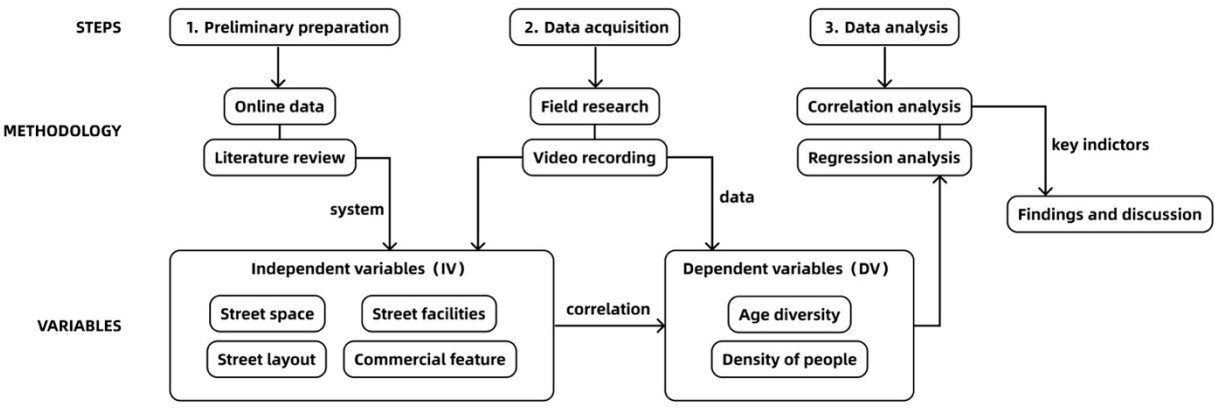

**Figure 1.** Research design (Source: authors).

### 3.1. Research Scope

This research examined Golden Street in Shanghai's congested Gubei area of Changning District. Golden Street is a compact yet refined community commercial road, spanning 700 m and ranging in width from 46 to 80 m. The street's design incorporates commercial and recreational spaces, leading to high foot traffic. Moreover, the area's residents are evenly distributed in terms of age, and the street offers a broad range of recreational activities. Figure 2 depicts street-view photographs of Golden Street.

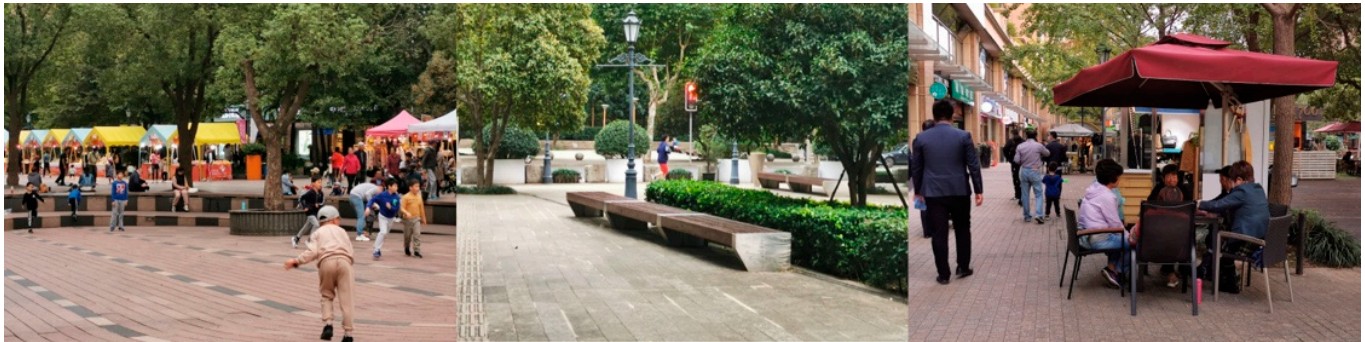

**Figure 2.** Views of Golden Street (Source: the authors).

### 3.2. Research Object

#### 3.2.1. Demographic Indicators

In 2016, the seven residential areas proximate to Golden Street had a combined population of 12,200 and exhibited an evenly distributed age composition. As a representative

thoroughfare of Shanghai, Golden Street had convenient transportation facilities, making it a compelling site for research inquiries. The primary activities observed on the street are walking, resting, exercising, and dining. The abundant foot traffic and diverse populace provided an optimal sample pool for this study. Our research noted two types of eclectic age convergence in our preliminary investigations: simultaneous and staggered sharing. Both scenarios indicated high age diversity, and optimising space utilisation via temporal segmentation was a viable strategy that did not compromise age diversity.

This research used field surveys, street scene video screening, data induction, and calculations to investigate Golden Street comprehensively. Following the age classification standard proposed by the WHO, our research divided the population into four categories: children (0–14 years old), young people (15–45 years old), middle-aged people (46–65 years old), and elderly people (over 65 years old). Based on video playback, researchers calculated the occurrence frequency of each group over four days, including both weekdays and weekends, across different periods. To facilitate ongoing research, the team installed a video recording device in October 2021, selecting weekdays with suitable weather conditions and avoiding holidays and weekends. The researchers recorded in groups from 8:00 a.m. to 6:00 p.m. every 30 min, generating a complete street walk video for each period. As a result of poor night vision and feedback from the survey results, pedestrian traffic after 6:00 p.m. was excluded from the survey scope. Moreover, observations at night were omitted because Golden Street had faint lighting and temporary shops in the square area, creating an unpredictable environment that could introduce randomness into the survey.

### 3.2.2. Spatial Indicators

Golden Street was a thriving commercial pedestrian walkway that was divided into three sections based on usage. The north and south ends of the street had a wide variety of commercial stores, and the central area featured a beautifully landscaped garden-style resting space. After an initial field investigation and brief data analysis, the distribution of the small-scale public spaces along Golden Street was constructed, and 50 small-scale public places were categorised based on commercial types, paved roads, seating arrangements, intersections, main roads, and branch roads [33,39,40]. All of the samples adhered to the principle of being open public spaces within the study area of the street and had varying sizes [23]. This approach ensured sufficient sample data for the analysis process, as demonstrated in Figure 3. Additionally, each place had a similar scope but differed in street composition components, which facilitated subsequent data analysis.

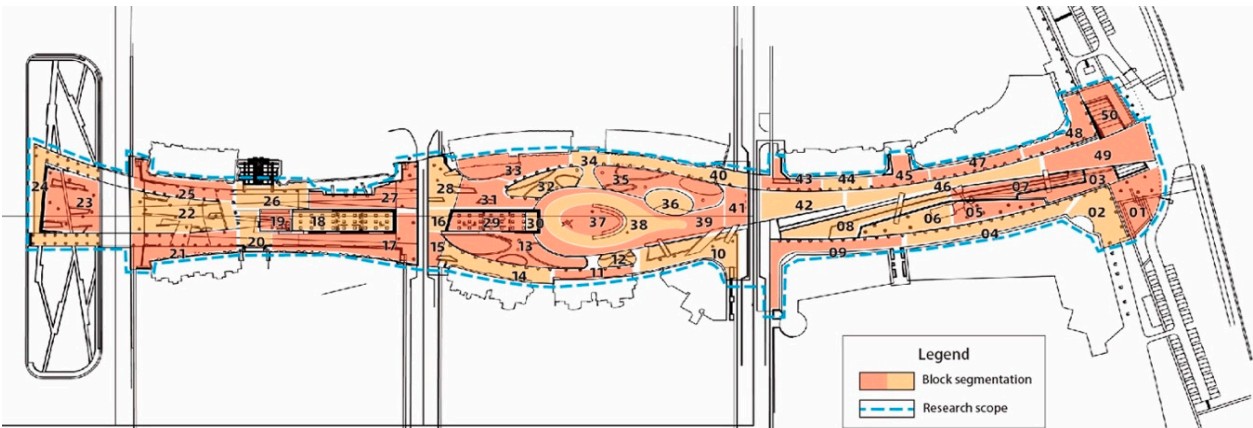

**Figure 3.** Map of Golden Street and subdivision plan (Source: the authors, the numbers in the picture mean the sample number in the research, sorted by location).

This research identified the independent variables as street components, comprising four aspects: street space, street facilities, street layout, and commercial features. Street space included the length and width of the street, and because Golden Street was a com-

mercial pedestrian walkway, facade transparency and tiling rate were integrated into the indicator system to measure the interaction and continuity of the facade. Facility components included the quantities of seats and trash cans, and an additional investigation was conducted into the density of amusement facilities such as combination slides and climbing facilities because of their abundance in the research area. Street layout included walking area proportion and street connectivity, and binary variables were used to define the commercial features [41].

Two indicators were used as dependent variables to measure the street crowd's vitality and assess the distribution of people by age. Crowd density reflected the change in the quantity of people [42], and age diversity indicated their quality. In combination, the two indicators provided feedback for determining the street's vitality index.

For example, sample No. 2 was located at the corner of Golden Street and the city's main road. The whole place was enclosed and rectangular in shape. Business was flourishing here, with many leisure and entertainment booths and a complete infrastructure. Due to its unique location, the street had a large flow of people and was close to the subway entrance, which was at the beginning of Golden Street.

*3.3. Research Hypothesis*

After conducting a preliminary investigation, researchers hypothesised the following: amusement facilities exhibit a positive correlation with age diversity, and infrastructure such as garbage cans, building features, and seating arrangements can also contribute to age diversity [15,36]; street width is inversely related to crowd density [16]; and business factors have a negative correlation with age diversity, as the elderly and children may have little interest in commercial activities. The influence of commercial features has been debated among scholars, with varying results reported [8,16]. This could be attributed to differences in business quality and regional variations in people's preferences and needs.

## 4. Methodology
### 4.1. Dependent Variables: Age Diversity and Crowd Density

1. The age diversity index (*AD*) is determined by applying Shannon's evenness index formula, which assesses the extent of age integration of individuals on the street [8,15,36]. This index indicates whether the street accommodates people of all ages, including the elderly and children. The index is determined as follows:

$$AD = \frac{\sum_{i=1}^{k} R_i[\ln(R_i)]}{\ln(k)} \tag{1}$$

Here, $k$ refers to the number of age categories in the street space, with $k = 4$. $R_i$ represents the proportion of individuals in age category $i$ to the total population, where 1 to 4 indicate adolescents, young people, middle-aged people, and elderly people, respectively. A value of 0 indicates a lack of diversity, and a value of 1 indicates a uniform distribution of age categories, that is, a completely homogeneous state. The Shannon evenness index ranges from 0 to 1, with values closer to 1 indicating greater age diversity, whereby each age group's demographic is evenly distributed and there is no dominant type present.

2. The crowd density index visually represents the number of individuals [43]. The formula used is as follows:

$$Density = \frac{n}{m} \tag{2}$$

where $n$ represents the total number of visits and $m$ represents the area of the divided small-scale public place in square metres. A higher density value indicates more individuals passing through and staying in the area, resulting in larger crowds. This suggests that the area is highly vibrant and that people gather here for numerous reasons. However, it is necessary to consider age diversity to analyse the completeness of all-age streets.

*4.2. Independent Variable: Living Street Components*

Using Golden Street as a research case, this research identified 4 primary and 10 secondary indicators applicable to the street through the literature review. According to the literature review, the index system increased the number of amusement facilities and reduced traffic accessibility, sky openness, and the green view rate because this part of the index is unsuitable for the Golden Street scenario. A binary variable was used for the commercial features. Various methods were used to collect data for each indicator. The following are the indicators and their quantification methods:

1. Street length: determined by measuring the length of the street centreline.

2. Street width [16]: calculated as the average width of the street.

3. Facade transparency [43]: determined by calculating the ratio of the transparent visible facade of a store to the total length of the street. The transparency data are classified based on the degree of spatial permeability of each store, which includes open storefronts with fully available frontage (A), transparent glass windows with a direct view of the interior (B), and advertising glass windows with product scenery (C). The formula for calculating transparency is (length of A interface × 1.25 + length of B interface + length of interface C × 0.75)/total length of the building interface along the street.

Note: Since public places do not contain the main body of a building, the elevation transparency of these public places was set at 1, indicating that these places have little impact on people's visual activities.

4. Facade sticker rate: length of building facade as a percentage of total public place length.

5. Seat density [39]: the number of seats in each public place divided into the two categories of formal seats and class seats, including the edge of the parterre and road edge class seats, to a social distance of 1.2 m accounting for the number of seats [44], thus set to seat density = (around the number of rattan chairs + the number of plastic seats + length of wooden benches/1.2).

6. Trash can density: the number of trash cans within each public place.

7. Amusement facility density: the number of amusement facilities within each public place.

8. Proportion of walking area [45]: the map indicates the percentage, set as flat pavement suitable for playing and walking areas.

9. Street connectivity [45,46]: the number of secondary spaces connected to each place to measure the degree of space openness and connectivity properties.

10. Commercial features [45,47]: Following the Chinese Community Commercial Development Specifications and Community Commercial Facilities Setting and Functional Requirements, this research classified and sorted the commercial forms along the street and then adjusted them according to the research area, which can be divided into 4 categories and 11 sub-categories, as shown in Table 3 below. Researchers used binary variables to calculate the number of business categories, and identified each category's data as an indicator.

The spatial index system and the various calculation methods used in this research site are shown in Table 4.

*4.3. Analytical Method*

Based on the small-scale public place division, researchers conducted a partition measurement to determine the relationships between the independent variables, including 10 indicators of street components, and the dependent variables, age diversity and crowd density. This research used SPSS.24 software to conduct Pearson correlation analyses to determine the correlations between the variables, then used linear regression to more deeply explore the relationship between the variables, analyse the logic of changes with the dependent variables, and identify the significant factors affecting the vitality of the street [48].

**Table 3.** Classification of business types.

| Major Categories | Minor Categories |
|---|---|
| Retail Shopping | Daily Use<br>Clothing Accessories<br>Specialty Stores<br>General Department Store |
| Catering | Mobile Booths<br>Catering |
| Leisure and Entertainment | Leisure and Entertainment<br>Education and Training<br>Family Services |
| Services | Medical Insurance<br>Financial Insurance |

**Table 4.** Summary of calculation methods of the independent variables.

| | Indicators | Calculation Mode |
|---|---|---|
| Street Space | Street length | Map survey |
| | Street width | Map survey |
| | Facade transparency | (Length of A interface × 1.25 + length of B interface length + length of interface C × 0.75)/total length of the building interface |
| | Facade sticker rate | Floor area/total street length |
| Street Facilities | Seat density | (Number of rattan chairs + number of plastic seats + length of wooden benches/1.2) |
| | Trash can density | Number of trash cans/place area |
| | Amusement facility density | Number of amusement facilities/place area |
| Street Layout | Proportion of walking area | Walking area/place area |
| | Street connectivity | Number of secondary spaces connected to each place |
| Commercial features | Commercial stores | Binary variable |

## 5. Findings

### 5.1. Analysis of Spatial Distribution Components of Street Crowd Density

According to the survey results, the crowd density index data showed several crowd-gathering points on the street, as shown in Figure 4. Samples No. 4 and 5 were significant centres where the commercial layout was dense, the seats were arranged in a reasonable way, and the field was open. People gathered here for trading, exercise, chess, and other activities, especially around 8:00 a.m. Elderly people in nearby communities gathered around the square to dance, and at around 4 p.m., many found a place to play chess. The square area in the centre of samples No. 37 and 38 was large and without shade, so crowds gathered at peak times, mainly to rest, play, and engage in other activities. Moreover, students skated near the square or socialised after school. At the same time, young people would come as parents, and the elderly would rest and take care of children.

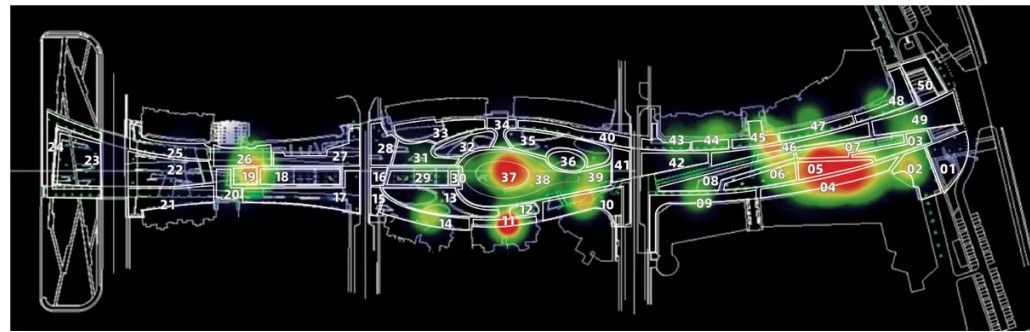

**Figure 4.** Spatial distribution of demographic density (Source: the authors, the numbers in the picture mean the sample number in the research).

### 5.2. Analysis of the Spatial Distribution Components of the Age Diversity of the Street Demographic

The spatial distribution of the age diversity indicators was roughly similar to that of the crowd density indicators. As shown in Figure 5, the age diversity indicators showed peak values and performed well in the central square area consisting of samples No. 37 and 38. The surrounding samples showed a paracentric development with better overall balance, followed by samples No. 5 and 6. Compared with the crowd density, the spatial distribution of age diversity was more uniform, especially between samples No. 20 and 27. The values in this area were generally low, but the distribution was scattered, and there was no strong dominance by a specific age group.

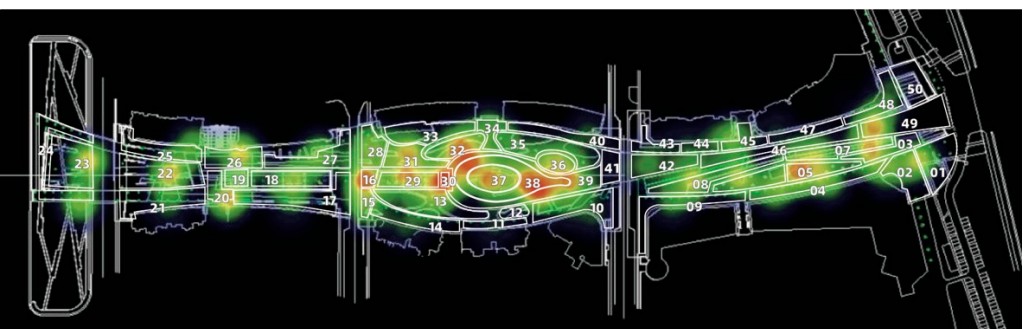

**Figure 5.** Spatial distribution of demographic age diversity (Source: the authors, the numbers in the picture mean the sample number in the research).

### 5.3. Correlation Analysis of Constituents Affecting Street Dynamics

5.3.1. Variable Correlation Test

The correlation analysis of the dependent variables crowd density, age diversity, and independent variables revealed that the following indicators were strongly related to crowd density: walking area, street width, seat density, and catering business. In addition, facade sticker rate, facade transparency, the number of amusement facilities, and leisure and entertainment businesses were strongly correlated with age diversity. The detailed results are shown in Table 5.

**Table 5.** Results of the correlation analysis.

|  | Crowd Density | Age Diversity |
|---|---|---|
| Proportion of Walking Area | 0.359 ** | 0.171 |
| Street Length | 0.095 | −0.059 |
| Street Width | −0.499 *** | 0.09 |
| Seat Density | 0.33 ** | −0.266 |
| Facade Sticker Rate | 0.239 | −0.475 *** |
| Facade Transparency | −0.063 | 0.326 ** |
| Trash Can Density | 0.122 | 0.123 |
| Street Connectivity | 0.232 | 0.026 |
| Number of Amusement Facilities | 0.069 | 0.421 *** |
| Retail Shopping | −0.168 | −0.13 |
| Catering | 0.385 *** | −0.109 |
| Leisure and Entertainment | 0.035 | −0.382 *** |
| Services | −0.036 | 0.185 |

** means $p < 0.05$; *** means $p < 0.01$.

### 5.3.2. Linear Regression Analysis

Further analysis was conducted of the correlation results using the linear regression method, and the results are shown in Tables 6 and 7 below:

**Table 6.** Linear regression analysis (crowd density correlation).

|  | Nonstandardised Coefficient | | Standardised Coefficient | $t$ | $p$ | Collinearity Diagnosis | |
|---|---|---|---|---|---|---|---|
|  | B | Standard Error | Beta | | | VIF | Tolerance |
| Constant | 0.113 | 0.080 | - | 1.413 | 0.165 | - | - |
| Proportion of Walking Area | 0.177 | 0.068 | 0.295 | 2.610 | 0.012 ** | 1.124 | 0.890 |
| Street Width | −0.006 | 0.002 | −0.331 | −2.875 | 0.006 *** | 1.168 | 0.856 |
| Seat Density | 0.001 | 0.001 | 0.257 | 2.332 | 0.024 ** | 1.068 | 0.936 |
| Catering | 0.129 | 0.040 | 0.347 | 3.231 | 0.002 *** | 1.016 | 0.984 |
| $R^2$ | | | 0.489 | | | | |
| $F$ | | | $F_{(4.45)} = 10.777, p = 0.000$ | | | | |
| D-W | | | 1.776 | | | | |

** means $p < 0.05$; *** means $p < 0.01$.

**Table 7.** Linear regression analysis (age diversity correlation).

|  | Nonstandardised Coefficient | | Standardised Coefficient | $t$ | $p$ | Collinearity Diagnosis | |
|---|---|---|---|---|---|---|---|
|  | B | Standard Error | Beta | | | VIF | Tolerance |
| Constant | 0.777 | 0.147 | - | 5.266 | 0.000 *** | - | - |
| Facade Sticker Rate | −0.206 | 0.104 | −0.407 | −1.989 | 0.053 | 3.282 | 0.305 |
| Facade Transparency | −0.016 | 0.137 | −0.023 | −0.115 | 0.909 | 3.139 | 0.319 |
| Number of Amusement Facilities | 0.162 | 0.066 | 0.296 | 2.467 | 0.018 ** | 1.128 | 0.886 |
| Leisure and Entertainment | −0.129 | 0.048 | −0.307 | −2.688 | 0.010 ** | 1.024 | 0.977 |
| $R^2$ | | | 0.427 | | | | |
| $F$ | | | $F_{(4.45)} = 8.395, p = 0.000$ | | | | |
| D-W | | | 1.717 | | | | |

** means $p < 0.05$; *** means $p < 0.01$.

To analyse the performance of the crowd density index, this research conducted a Pearson correlation analysis, and multiple factors were shown to be prominent. As shown in Table 5, the model $R^2$ value of 0.489 implies that proportions of walking area, street width, seat density, and catering category can explain 48.9% of the variation in crowd density. The model passed the F-test (F = 10.777, $p = 0.000 < 0.05$), which means that at least one of these indicators affects crowd density. The D-W value is around 2, thus indicating that the model is not autocorrelated, there is no correlation between the sample data, and the model is good. Therefore, the formula for crowd density can be summarised as follows:

y = 0.113 + 0.177 * proportion of walking area −0.006 * street width +0.001 * seat density + 0.129 * catering.

To summarise the analysis, the proportion of walking area, seat density, and catering significantly affect crowd density. Street width also has a significant adverse effect on crowd density. These findings are consistent with our hypotheses.

In terms of effects on age diversity, the Pearson correlation results revealed factors such as amusement facilities and chair density to be prominent. The R-squared value of the model is 0.427, which implies that the façade alignment rate, facade transparency, number of amusement facilities, and leisure and entertainment businesses can explain 42.7% of the variation in age diversity. The model passed the F-test (F = 8.395, *p* = 0.000 < 0.05), which means that at least one of these indicators affects age diversity. There is no autocorrelation in the model and no correlation between the sample data, so the model is good. The formula is y = 0.777–0.206 * facade sticker rate −0.016 * facade transparency + 0.162 * number of amusement facilities −0.129 * leisure and entertainment.

Summarising the analysis, the number of amusement facilities and leisure and entertainment activities significantly negatively affects age diversity. However, the façade sticker ratfaçadefacade transparency do not have any influence on age diversity. The role of amusement facilities is consistent with our hypotheses, but the other factors do not play as vital a role as expected.

## 6. Discussion

### 6.1. Research Summary

In this study, researchers identified components that strongly correlated with the age diversity of street spaces. The findings of this research provided valuable insights into the creation of all-age-friendly environments. The following is a summary of the research outcomes and contributions in three distinct areas: age diversity, spatial components of streets, and management and policy-making.

### 6.1.1. Explanations of Age Diversity

The results of this study showed that the age diversity indicator could provide a reliable measure of the presence of individuals of all ages in a given space. Furthermore, correlation analysis with spatial components enabled the identification of critical indicators of the presence of all ages in improved areas [49].

Concerning infrastructure, amusement facilities, and seating, they could stimulate the presence of people of all ages in an area. Facade factors could also impact different age groups, and other types of businesses could attract diverse populations. Enhancing these factors could significantly improve the all-age friendliness of a street [8]. Moreover, the two primary areas of high age diversity on the street (samples No. 37–39 and 2–6) correspond to two forms of expression. Samples No. 37–39 exhibited high age diversity and crowd activity simultaneously, whereas samples No. 2–6 were active at different times.

This research had established an index system and easy-to-implement methods for analysing the age of streets, which could be replicated in similar open public spaces such as walking streets, parks, and communities. Follow-up research can use similar empirical studies to gather data on population counts, thereby providing a valuable reference for overall planning and renovation.

### 6.1.2. Suggestions for Spatial Components

First, commercial features are an important factor in driving age diversity and attracting foot traffic. This study's hypothesis regarding business factors was correct, as business factors showed a negative correlation with age diversity [15,30]. Samples No. 17 to 27 were situated too far from the main street and had poor layout, resulting in low pedestrian traffic. In contrast, samples No. 47 and 48 had unique store designs and components that drew more people and boosted the street's activity. To optimise commercial features and match them with different age groups, future investment should aim to diversify commercial

formats, introduce restaurants, entertainment, and parent–child activities, and modify small-scale public places with a lower commercial density [16,18]. Moreover, the presence of relatively open stores could extend people's stopping time, and any interesting feature on the street would stimulate social interaction among passers-by [50–53].

Consistent with our hypothesis, the amusement facilities, such as the children's slides and seesaws on Golden Street, were positively correlated with age diversity. Amusement facilities were helpful for age diversity, as previous findings show [52,53]. However, most adults only observed from the side lines and used their mobile devices or occupied the children's amusement facilities while children were playing [27,34]. In addition, the age diversity of samples No. 18 and 29 with amusement facilities was different, which was related to the effect of other factors. For future planning, it might be more advantageous to include entertainment facilities for young people and the elderly that suit their preferences, as well as rest areas or adult fitness equipment.

Finally, concerning street facilities, apart from the seating factor, the street factors (such as garbage cans) did not have a significant correlation with age diversity. This research found that rest seats had a high utilisation rate [30] and were helpful for all age groups. These conclusions are related to the findings of Paydar [39], Main [54], and Shaftoe [55]. Elderly and young people often use benches while watching children play. Hence, more diverse and reasonable seating arrangements could improve resident communication and increase street vitality.

### 6.1.3. Insights for Management and Policy-Making

During the initial project development and transformation stage, developers could conduct field research to gauge people's preferences for street usage based on age diversity. These data could then be utilised in the design stage to make informed decisions on critical elements that require improvement. Commerce, infrastructure, and spatial layout could be improved to create a positive neighbourhood atmosphere that was all-age-friendly. By focusing on critical factors related to age diversity, such as all-age sharing, targeted improvement measures could be implemented to enhance street planning.

For instance, on Golden Street, small-scale public places had been constructed to be all-age-friendly. Similar construction strategies had been implemented in small and micro spaces elsewhere to enhance residents' quality of life. By applying this research method and index system to wider streets or communities, there will be more interesting findings and constructive suggestions.

### 6.2. Limitations and Prospects for Urban Street Development

This study had several limitations that could be briefly summarised as follows: (1) The data analysed in this study were obtained from an area of limited size that may not be representative of every street or city because of the unique factors that can influence age diversity. (2) The correlation results obtained from the data analysis were not entirely satisfactory, and other overlapping factors may have influenced age diversity. (3) The volume of observational data used in this study was insufficient, as the level of activity and capacity of a street with high ornamental value could vary significantly across different seasons or years, and the effects of time, weather [56], building walls, and trees may also be important.

In conclusion, this research constructed age diversity and linked it to street components, providing practical insights for the future planning of small-scale spaces and street public life. Emphasising small and micro public spaces in current planning and design would help shape the 15-min community-life circle, and considering the distribution of age groups in small spaces could also promote human-centred transformation. Through gradual micro-renewal, small-scale spaces could be integrated into large-scale development, improving residents' daily lives and fostering coexistence between generations. This represented a new direction for urban street development.

**Author Contributions:** Methodology, Y.L.; Writing—original draft, Y.L.; Writing—review & editing, D.A. and Y.H.; Visualization, Y.H.; Project administration, D.A. All authors have read and agreed to the published version of the manuscript.

**Funding:** This work is supported by the Youth Foundation of Humanities and Social Sciences of the Ministry of Education in China under Grants 21YJC760031 and 22YJC760001.

**Institutional Review Board Statement:** Not applicable.

**Informed Consent Statement:** Not applicable.

**Data Availability Statement:** Some or all of the data, models or codes supporting the results of this study may be obtained from the respective authors.

**Conflicts of Interest:** The authors declare no conflict of interest.

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
