# Peer review of "The Influence of Street Components on Age Diversity: A Case Study on a Living Street in Shanghai"

_sustainability, doi:10.3390/su151310493_

Round 1
Reviewer 1 Report
The presentation of the work is confusing and ill-structured. The objectives of the work must be clear and be at the beginning of the work. Badly numbered chapters. Badly numbered references. Not write in the first person singular or in the third person plural. Misnumbered figures, starting at figure 1-3 without showing 1-1 and 1-2. Cite figures in the text, numbering the equations and referencing. Table 3 is missing. Font size issues in section 4.3.1
Improve the quality of english
Author Response
Response to Reviewer Comments
Dear reviewer,
Re: Manuscript ID: sustainability-2411912 and Title: The Influence of Street Components on Age Diversity: A Case Study on A Living Street in Shanghai.
Thank you for your letter and for commenting as a reviewer on our manuscript. Those comments are all valuable and helpful for revising and improving our paper and the essential guiding significance of our research. We have studied the comments carefully and have made corrections which we hope meet with approval. The modified part is marked in MS Word using the revised mode. Meanwhile, in the attachment, we provide the revised and clean versions for your review. The corrections in the paper and the response to the reviewer's comments are as follows:
Responds to the reviewer's comments:
Point 1: The presentation of the work is confusing and ill-structured.
Response 1: Thank you for your suggestions, and sorry for not carefully checking the structure and logic of the article in the initial manuscript submission and disturbing your review due to the misplaced formatting. We have compiled the entire text in the hope that you will approve our corrections.
Point 2: The objectives of the work must be clear and be at the beginning of the work.
Response 2: Thank you sincerely for your valuable suggestions. We have reorganized the part of the work objectives (revised-Line 129, clean-Line 52), put forward and clarified our research objectives as soon as possible after the background introduction of the first part, and in 1. The introduction corresponds to the research objectives parts, while the conclusion is strengthened to echo the objectives mentioned at the beginning, making the article's idea more evident.
Point 3: Badly numbered chapters. Badly numbered references.
Response 3: We apologize for our careless mistakes and have readjusted the relevant chapter numbering and literature citations.
Point 4: Not write in the first person singular or in the third person plural.
Response 4: We apologize for our careless mistakes and grammatical errors. We have checked and corrected the full text.
Point 5: Misnumbered figures, starting at figure 1-3 without showing 1-1 and 1-2.
Response 5: Sorry for this error. We meant to express images 1, 2, and 3, but this statement is ambiguous, so we have renumbered the chart according to your suggestion.
Point 6: Cite figures in the text, numbering the equations and referencing.
Response 6: Sorry for this error. The image part is self-drawn, and we have marked the source, formulas, and references citations we have readjusted.
Point 7: Table 3 is missing.
Response 7: Sorry for this error. We have readjusted the chart order and numbering.
Point 8: Font size issues in section 4.3.1
Response 8: Sorry for this error. We've readjusted the text style issue for the full text.
Point 9: Comments on the Quality of English Language
Response 9: Thanks for your suggestion. I searched for a native English speaker to re-polish the text, and we hope the revised manuscript could be acceptable to you.
We tried our best to improve the manuscript and made some changes. We appreciate the editors and reviewers for their enthusiastic work earnestly and hope the correction will be approved. Once again, thank you very much for your comments and suggestions. By the way, I have uploaded the new manuscript in the attachment.
Yours sincerely,
Yan Liu

Reviewer 2 Report
The author announces a broad review of the literature, justifying the formulation of the index system for the purposes of the study. Literature research has certainly been carried out. This is evidenced by the extensive tables 1 and 2 (wrongly formatted and therefore largely illegible). More consideration should be given to the information collected in the tables in the literature review. In particular, the text in part one does not follow the three main methodological conclusions formulated in section 1.5. They should be strengthened on the basis of the literature review.
In the study, despite statistical analyses, no research hypotheses were formulated, which results in the impression of random selection of the explained results. Apart from the research objectives (2.3), research hypotheses should be formulated.
As mentioned, the results of the study - however interesting they seem, are disordered. The description should be structured. There is also a need to expand the discussion. Currently, these are loose notes summarizing the results, not a discussion with the literature cited in the review. No conclusions were presented at all.
Need proofreading.
Also, footnotes have been edited incorrectly - Oxford style is mixed with Harvard style.
Author Response
Response to Reviewer Comments
Dear reviewer,
Re: Manuscript ID: sustainability-2411912 and Title: The Influence of Street Components on Age Diversity: A Case Study on A Living Street in Shanghai.
Thank you for your letter and for commenting as a reviewer on our manuscript. Those comments are all valuable and helpful for revising and improving our paper and the essential guiding significance of our research. We have studied the comments carefully and have made corrections which we hope meet with approval. Revised portions are marked in red on the paper. The corrections in the paper and the response to the reviewer's comments are as follows:
Responds to the reviewer's comments:
Point 1: The author announces a broad review of the literature, justifying the formulation of the index system for the purposes of the study. Literature research has certainly been carried out. This is evidenced by the extensive tables 1 and 2 (wrongly formatted and therefore largely illegible). More consideration should be given to the information collected in the tables in the literature review. In particular, the text in part one does not follow the three main methodological conclusions formulated in section 1.5. They should be strengthened on the basis of the literature review.
Response 1: We sincerely thank you for your valuable suggestions. First of all, we are very sorry about the format of Table 1-2, this part has been adjusted in the new manuscript, and the content of Table 1-2 is related to representative literature at home and abroad in the field of research, by combing we will divide the content of the table into four parts, respectively 2.1 to 2.4. By analyzing the literature, we hope to guide the index system and methods of this study, and we also refer to your suggestions to adjust the literature review part. The four components of 2.1 to 2.4 add the interpretation and literature citation of Table 1-2, reconstruct the review part, and emphasize at the beginning of the literature review that this paper focuses on quantitative experiments on all-age indicators rather than concept research, so the review part focuses on highlighting methods and logic, combing and summarizing the relevant literature on age indicators, and adjusting the summary in 2.5 to correspond to the review conclusions, hoping that you can recognize our correction.
Point 2: In the study, despite statistical analyses, no research hypotheses were formulated, which results in the impression of random selection of the explained results. Apart from the research objectives (2.3), research hypotheses should be formulated.
Response 2: Thank you sincerely for your valuable advice. The lack of research hypotheses is our negligence. We formulated the hypotheses of this research based on the literature at the end of the study design (3.4: revised-Line 912, clean-Line 267) and confirmed the hypotheses according to the analysis results at the conclusion stage, and indeed found interesting comparisons, some conclusions are different from the existing literature research, I hope you can recognize our correction.
Point 3: As mentioned, the results of the study - however interesting they seem, are disordered. The description should be structured. There is also a need to expand the discussion. Currently, these are loose notes summarizing the results, not a discussion with the literature cited in the review. No conclusions were presented at all.
Response 3: Thank you for your suggestion. In the conclusion part, we have reorganized three parts for all-age indicators, street space indicators, and policy-making, corresponding to the structure of the data analysis part while expanding the scope of discussion, this research guides to a broader range of interests and practices, the collection and analysis methods of age indicators can be used on a larger scale, and the correlation of street spatial components also corresponds to the improvement direction of policy-making; Meanwhile, the conclusions are compared and analyzed with the existing literature, the key influencing indicators of the case site are analyzed and elaborated, and the improvement plan is proposed in a targeted manner, hoping that you can recognize our correction.
Point 4: Comments on the Quality of English Language
Response 4: Thanks for your suggestion. I searched for a native English speaker and re-edited the whole text. The footnotes have been modified to follow the Sustainability format. And we hope the revised manuscript could be acceptable to you.
We tried our best to improve the manuscript and made some changes. We appreciate the editors and reviewers for their enthusiastic work earnestly and hope the correction will be approved. Once again, thank you very much for your comments and suggestions. By the way, I have uploaded the new manuscript in the attachment.
Yours sincerely,
Yan Liu

Reviewer 3 Report
General Suggestions:
• The author should provide a clearer definition of "age-friendly" or "age-diversity" street environments. Currently, the manuscript suggests that age diversity refers to different age groups using the same street simultaneously, but it does not discuss whether a street accommodating different age groups at different times can also be considered age-friendly. Is a street used by different age groups at different times not considered an "age-diverse" street or an "age-friendly" street? A clearer definition and discussion need to be specified early in the manuscript.
• Elaborate on the contribution of this case study within a larger context. As the study is specific to a particular context, explain how it can generate research interest among a broader audience.
Major questions:
• In the Review section, the emphasis is primarily on procedural elements and methodologies, while there is limited discussion on existing literature that addresses street characteristics in relation to different age demographics. It would be academically more robust to establish the research gap from a theoretical perspective.
• Line, 195, Provide further details on the rationale behind selecting 50 blocks, including the decision-making process for choosing this specific number. The current explanation is too generalized.
•Line 438: Expand on why the "enclosed seating" layout would be preferred, and reference the specific findings that support this preference.
Minor questions:
• Ensure that the formatting of Table 1 and Table 2 remains intact. Including sub-section division lines could be helpful for readers' comprehension.
• Make sure the zone numbers in Figure 4 are large enough to read clearly. Currently, they appear unclear and too small.
• Check lines 219 to 223 to avoid repetition of bullet points.
• Consider including block divisions on Figure 5 to facilitate readers' understanding and follow the result discussion.
• Address formatting issues in Table 7, such as ensuring the table remains within the page and maintaining consistent table borderline weights.
• Apply the same formatting adjustments to Table 8 as for Table 7.
Minor
Author Response
Response to Reviewer Comments
Dear reviewer,
Re: Manuscript ID: sustainability-2411912 and Title: The Influence of Street Components on Age Diversity: A Case Study on A Living Street in Shanghai.
Thank you for your letter and for commenting as a reviewer on our manuscript. Those comments are all valuable and helpful for revising and improving our paper and the essential guiding significance of our research. We have studied the comments carefully and have made corrections which we hope meet with approval. Revised portions are marked in red on the paper. The corrections in the paper and the response to the reviewer's comments are as follows:
Responds to the reviewer's comments:
General Suggestions:
Point 1: The author should provide a clearer definition of "age-friendly" or "age-diversity" street environments. Currently, the manuscript suggests that age diversity refers to different age groups using the same street simultaneously, but it does not discuss whether a street accommodating different age groups at different times can also be considered age-friendly. Is a street used by different age groups at different times not considered an "age-diverse" street or an "age-friendly" street? A clearer definition and discussion need to be specified early in the manuscript.
Response 1: We sincerely thank you for your valuable suggestions. We have briefly discussed the definition of all-age friendliness in the conclusion section, but according to your suggestions, we also think that it is necessary to summarize in the early stage of the article, in fact, two kinds of "all-age friendly" that is, both contemporary and time-sharing, are the embodiment of all-age friendliness. In the quantitative research, the definition of the two forms is not distinguished because time-sharing activities can promote intergenerational communication, and contemporary activities can improve space utilization, laying the foundation for creating flexible space. Both are beneficial for all ages. We have added more elaboration to this in the starter manuscript's introduction and literature review sections (revised-Line 365, clean-Line 141). And we hope the revised manuscript could be acceptable to you.
Point 2: Elaborate on the contribution of this case study within a larger context. As the study is specific to a particular context, explain how it can generate research interest among a broader audience.
Response 2: Thank you for your suggestion. We have reorganized the theoretical application scenarios in the conclusion part. At the beginning of the conclusion, we emphasized the broader application scope of this research, organized into three parts for all-age indicators, street space indicators, and policy-making, corresponding to the structure of the data analysis part, expand the scope of discussion, and guide the research to a broader range of interests and practices, due to the reproducibility of the index system and experimental methods, it is also feasible for the same type of area, and then in different contexts, such as parks. Similar empirical studies can conduct in the pedestrian street. We hope that you can recognize our corrections.
Major questions:
Point 3: In the Review section, the emphasis is primarily on procedural elements and methodologies, while there is limited discussion on existing literature that addresses street characteristics in relation to different age demographics. It would be academically more robust to establish the research gap from a theoretical perspective.
Response 3: We sincerely thank you for your valuable suggestions. We have adjusted the literature review section concerning your suggestions, reconstructed the review section, and emphasized at the beginning of the literature review that this paper focuses on quantitative experiments on all-age indicators rather than conceptual research. Hence, the review focuses on methods and logic (including the content of Tables 1-2), and you suggest discussing street characteristics and studies between different age groups. We also found relevant literature and cited the discussion (8 -9), but this part only discusses the differences between various groups. In addition, some directions only concern the elderly or children. It does not integrate considerations, so this paper uses age diversity as a confluent indicator. We hope that you can recognize our correction.
Point 4: Line, 195, Provide further details on the rationale behind selecting 50 blocks, including the decision-making process for choosing this specific number. The current explanation is too generalized.
Response 4: Regarding the division of blocks, we refer to the previous literature (52-53) according to the specific situation of the case site divided (revised-Line 832, clean-Line 239). To achieve statistical significance and meet the experiment's needs, we get the number 50 after partitioning, which has also been updated in the manuscript. We hope that you can recognize our correction.
Point 5: Line 438: Expand on why the "enclosed seating" layout would be preferred, and reference the specific findings that support this preference.
Response 5: The conclusion of the seat part has been structured and adjusted, and the conclusion of the enclosed seat is not much related to the research in this paper, so the elaboration of this part has been simplified but supplemented by other literature on seat performance.
Minor questions:
Point 6: Ensure that the formatting of Table 1 and Table 2 remains intact. Including sub-section division lines could be helpful for readers' comprehension.
Response 6: We are very sorry about the formatting of Tables 1-2. This part has been adjusted in the new manuscript, sorted out the formatting problems of the three-line table, and we hope you will recognize our correction.
Point 7: Make sure the zone numbers in Figure 4 are large enough to read clearly. Currently, they appear unclear and too small.
Response 7: We are very sorry for the problem with the clarity of the pictures. This section has been adjusted in the new manuscript, the digital display and clarity have been organized, and we hope you will recognize our correction.
Point 8: Check lines 219 to 223 to avoid repetition of bullet points.
Response 8: We are very sorry for the problems caused by our carelessness. This part has been adjusted in the new manuscript, and we hope you will recognize our correction.
Point 9: Consider including block divisions on Figure 5 to facilitate readers' understanding and follow the result discussion.
Response 9: Thank you for your sincere suggestion. We have corrected it so it helps to read and understand. I hope you can recognize our correction.
Point 10: Address formatting issues in Table 7, such as ensuring the table remains within the page and maintaining consistent table borderline weights.
Response 10: We are very sorry for the problems caused by our carelessness. This part has been adjusted in the new manuscript, and we hope you will recognize our correction.
Point 11: Apply the same formatting adjustments to Table 8 as for Table 7.
Response 11: We are very sorry for the problems caused by our carelessness. Due to the adjustment of the article's structure, the significance of the discussion between characteristic spatial indicators is not apparent and insufficient, so we have removed Table 8, hoping that you can recognize our correction.
Point 12: Comments on the Quality of English Language: Minor
Response 12: Thanks for your suggestion. I searched for a native English speaker and re-edited the whole text. And we hope the revised manuscript could be acceptable to you.
We tried our best to improve the manuscript and made some changes. We appreciate the editors and reviewers for their enthusiastic work earnestly and hope the correction will be approved. Once again, thank you very much for your comments and suggestions. By the way, I have uploaded the new manuscript in the attachment.
Yours sincerely,
Yan Liu

Reviewer 4 Report
Review of sustainability-2411912
: "The Influence of Street Components on Age Diversity: A Case Study on A Living Street in Shanghai : Dadi An and Yan Li
Recommendation: Minor Revision
Comments to the Author
The manuscript presents a strong correlation between street facilities and age diversity, which was not included in previous research. In addition, different domains have street space, street facilities, street layouts, and commercial features. So this paper can significantly contribute to living streets for other age groups. The discussion is well-structured, with findings emphasized. I recommend it for publication in sustainability. However, minor comments about role of weather should be addressed before accepting this manuscript.
I recommend the publication of this manuscript after minor revisions in light of the comments below.
Major comments:
This paper has discussed street elements and linked them to age diversity, providing practical insights for future planning of small-scale spaces and street public life. Therefore the role of weather, including wall, road temperature effects, radiation emissivity effects, and tree effects, needs to include.
Minor comments
[L-218] Please check the repeat numbers 1, 2, and 3.
[L-231] Please re-write I don't understand the equation
[L-306] time should be a.m.?
Tables 6 and 7. Please maintain the consistency of significant digits throughout the paper
[L-394] Please maintain the consistency of time PM or P.M.
Please check the equation and variables. Many places need clarification. The equation variable could be more defined. Proofreading the manuscript will be beneficial.
Author Response
Response to Reviewer Comments
Dear reviewer,
Re: Manuscript ID: sustainability-2411912 and Title: The Influence of Street Components on Age Diversity: A Case Study on A Living Street in Shanghai.
Thank you for your letter and for commenting as a reviewer on our manuscript. Those comments are all valuable and very helpful for revising and improving our paper, as well as the essential guiding significance to our research. We have studied the comments carefully and have made corrections which we hope meet with approval. Revised portions are marked in red on the paper. The corrections in the paper and the response to the reviewer's comments are as follows:
Responds to the reviewer's comments:
Major comments:
Point 1: This paper has discussed street elements and linked them to age diversity, providing practical insights for future planning of small-scale spaces and street public life. Therefore, the role of weather, including wall, road temperature effects, radiation emissivity effects, and tree effects, needs to include.
Response 1: Thank you for your valuable advice. These factors were also considered in the early stages of our experiment. Thank you very much for your help, which expands our thinking about subsequent investigations and discusses back to this article. We put these future thoughts into limitations and shortcomings and hope that you can recognize our correction.
Minor comments
Point 2: [L-218] Please check the repeat numbers 1, 2, and 3.
Response 2: We apologize for the problems caused by our carelessness, but we have reverted this part in the new manuscript.
Point 3: [L-231] Please re-write I don't understand the equation
Response 3: We are very sorry for the problem caused by our carelessness. We have rewritten all the formulas in the new manuscript and added an explanation section, hoping you will recognize our correction.
Point 4: [L-306] time should be a.m.?
Response 4: We apologize for the problems caused by our carelessness, but we have reverted this part in the new manuscript.
Point 5: Tables 6 and 7. Please maintain the consistency of significant digits throughout the paper
Response 5: We are very sorry for the formatting problem caused by our carelessness. We have corrected this part in the new manuscript and the relevant data interpretation: Table 6 is the correlation analysis; we change the context of Table 7 and Table 8 into the linear regression data. According to the literature review, we chose to combine these two methods to explore the correlation, abandoning stepwise regression because the method has been proven to be flawed by some scholars, and the critical indicators obtained by linear regression coincide with the correlation, their numerical analysis results are progressive, and such a method is more suitable for this study.
Point 6: [L-394] Please maintain the consistency of time PM or P.M.
Response 6: We apologize for the problems caused by our carelessness, but we have reverted this part in the new manuscript.
Point 7: Comments on the Quality of English Language: Please check the equation and variables. Many places need clarification. The equation variable could be more defined. Proofreading the manuscript will be beneficial.
Response 7: Thanks for your suggestion. We have fixed the formula problem, and I found a native English speaker to re-polish the whole text. And we hope the revised manuscript could be acceptable to you.
We tried our best to improve the manuscript and made some changes. We appreciate the editors and reviewers for their enthusiastic work earnestly and hope the correction will be approved. Once again, thank you very much for your comments and suggestions. By the way, I have uploaded the new manuscript in the attachment.
Yours sincerely,
Yan Liu

Round 2
Reviewer 1 Report
Please see the corrections included in the attached file

Moderate editing of English language required
Author Response
Response to Reviewer Comments
Dear reviewer,
Re: Manuscript ID: sustainability-2411912 and Title: The Influence of Street Components on Age Diversity: A Case Study on A Living Street in Shanghai.
Thank you for your letter and for commenting as a reviewer on our manuscript. Those comments are all valuable and helpful for revising and improving our paper and the essential guiding significance of our research. We have studied the comments carefully and have made corrections which we hope meet with approval. The modified part is marked in MS Word using the revised mode. Meanwhile, in the attachment, we provide the revised and clean versions for your review. The corrections in the paper and the response to the reviewer's comments are as follows:
Responds to the reviewer's comments:
Point 1: Comments marked in the attached file.
Response 1: First of all, thank you very much for your care and patience, we have read your suggestions carefully through the attached pdf file, but for some reason, actually we have corrected the formatting issues you raised in the first round of review (e.g., formatting issues with images, references, etc.) . However, the comment file we received in the second round is still the original version of the manuscript, so those formatting issues still exist. This may be due to network or file errors, as other reviewers did not have similar problems, so combining your two rounds of suggestions, we have corrected the errors in images, text, citations, etc. in the full text, and we hope the revised manuscript could be acceptable to you.
In addition, regarding language issues, basic grammatical errors have been fixed and we have found a native English speaker through AsiaEdit to proofread the article, so hopefully we can get your approval.
We tried our best to improve the manuscript and made some changes. We appreciate the editors and reviewers for their enthusiastic work earnestly and hope the correction will be approved. Once again, thank you very much for your comments and suggestions.
Yours sincerely,
Yan Liu

Reviewer 2 Report
The article has been largely rebuilt and supplemented with the issues indicated in the review. The methodological aspect has been particularly refined. In the current version, the text is eligible for publication after proofreading.
It is necessary to carefully read the text (proofreading), because due to numerous corrections in the text overwritten with the original text, there are stylistic inaccuracies.
Author Response
Response to Reviewer Comments
Dear reviewers,
Re: Manuscript ID: sustainability-2411912 and Title: The Influence of Street Components on Age Diversity: A Case Study on A Living Street in Shanghai.
Thank you for your letter and for commenting as a reviewer on our manuscript. Those comments are all valuable and helpful for revising and improving our paper and the essential guiding significance of our research. We have studied the comments carefully and have made corrections which we hope meet with approval. The modified part is marked in MS Word using the revised mode. Meanwhile, in the attachment, we provide the revised and clean versions for your review. The corrections in the paper and the response to the reviewer's comments are as follows:
Responds to the reviewer's comments:
Point 1: It is necessary to carefully read the text (proofreading), because due to numerous corrections in the text overwritten with the original text, there are stylistic inaccuracies.
Response 1: Thank you for your suggestions, for the full text order and textual issues have been sorted out, including tense, unification and other issues, we hope the revised manuscript could be acceptable to you. And thank you for your valuable suggestions in the first round, they were all very useful to me.
We tried our best to improve the manuscript and made some changes. We appreciate the editors and reviewers for their enthusiastic work earnestly and hope the correction will be approved. Once again, thank you very much for your comments and suggestions.
Yours sincerely,
Yan Liu

Reviewer 3 Report
no further comments
Author Response
Response to Reviewer Comments
Dear reviewers,
Re: Manuscript ID: sustainability-2411912 and Title: The Influence of Street Components on Age Diversity: A Case Study on A Living Street in Shanghai.
Thank you for your recognition, the suggestions you made in the first round were very useful, which helped this article to be further amended, thank you again for your hard work.
Yours sincerely,
Yan Liu
